# ASV vs OTUs clustering: Effects on alpha, beta, and gamma diversities in microbiome metabarcoding studies

**Andrea Fasolo, Saptarathi Deb, Piergiorgio Stevanato, Giuseppe Concheri, Andrea Squartini**(ORCID)*

Department of Agronomy, Animals, Food, Natural Resources and Environment DAFNAE, University of Padova, Padua, Italy

* squart@unipd.it

## Abstract

In microbial community sequencing, involving bacterial ribosomal 16S rDNA or fungal ITS, the targeted genes are the basis for taxonomical assignment. The traditional bioinformatical procedure has for decades made use of a clustering protocol by which sequences are pooled into packages of shared percent identity, typically at 97%, to yield Operational Technical Units (OTUs). Progress in the data processing methods has however led to the possibility of minimizing technical sequencers errors, which were the main reason for the OTU choice, and to analyze instead the exact Amplicon Sequence Variants (ASV) which is a choice yielding much less agglomerated reads. We have tested the two procedures on the same 16S metabarcoded bacterial amplicons dataset encompassing a series of samples from 17 adjacent habitats, taken across a 700 meter-long transect of different ecological conditions unfolding in a gradient spanning from cropland, through meadows, forest and all successional transitions up to the seashore, within the same coastal area. This design allowed to scan a high biodiversity basin and to measure alpha, beta and gamma diversity of the area, to verify the effect of the bioinformatics on the same data as concerns the values of ten different ecological indexes and other parameters. Two levels of progressive OTUs clustering, (99% and 97%) were compared with the ASV data. The results showed that the OTUs clustering proportionally led to a marked underestimation of the ecological indicators values for species diversity and to a distorted behaviour of the dominance and evenness indexes with respect to the direct use of the ASV data. Multivariate ordination analyses resulted also sensitive in terms of tree topology and coherence. Overall, data support the view that reference-based OTU clustering carries several misleading disadvantageous biases, including the risk of missing novel taxa which are yet unreferenced in databases. Since its alternatives as *de novo* clustering have on the other hand drawbacks due to heavier computational demand and results comparability, especially for environmental studies which contain several yet uncharacterized species, the direct ASV based analysis, at least for prokaryotes, appears to warrant significand advantages in comparison to OTU clustering at every level of percent identity cutoff.

**Data Availability Statement:** The sequence data were deposited in the NCBI Sequence Reads Archive (SRA) under code PRJNA608631.

**Funding:** Co-author G.C. was the recipient of a national grant from the University of Padova: "Progetto di Ateneo PRAT CPDA154841/15" The funders had no role in study design, data collection and analysis, decision to publish, or preparation of the manuscript.

**Competing interests:** The authors have declared that no competing interests exist.

## Introduction

The stages of development of high-throughput and Next Generation Sequencing methods have unfolded at such a fast pace in the past two decades, that nowadays the term NGS itself is sounding obsolete. Within the *in silico* methods the processing of the raw data has seen an important shift as regards the increasingly acknowledged value of handling the reads as amplified single variants (ASVs) to infer directly their taxonomy [1, 2] without clustering them first into Operational Technical Units (OTUs) packages of shared 97% homology [3, 4]. The 97% threshold had been originally chosen as it approximates the species cutoff homology boundary [5]. But the advancement in sequence denoising steps [6] has enabled a minimization of the sequencing errors which was the main reason for which the clustering of sequencing reads into OTUs had been originally adopted. In the early stages of the metagenomics era, the use of alignment algorithms against a known reference template was hampered by the risk that, even a limited number of single nucleotide variants due to the background sequencer error in base calling, would confound an aligner and cause mistaken final attributions. Such issue was felt particularly relevant in targeted sequencing (e.g. aiming at 16S meta barcodes) when working with community DNA from unknown environments, in which the focus is the comparison of multiple similar sequences, as opposed to other approaches as the alignments across multiple genomes of single isolates of certain origin. The metabarcoding context was thus prone to mis-attribution of a given sequence, causing either the false detection of a close, but incorrect taxon, or the false discovery of a new one. The OTU clustering strategy was initially the work-able walkaround to circumvent this potential bias. The clustering rationale rests upon the idea that related microorganisms have similar target gene sequences, over which, rare sequencing errors would have a negligible contribution to a given consensus sequence, whose cutoff could be arbitrarily imposed by grouping reads into operating taxonomic units (OTUs) sets [7].

Nevertheless, generating OTUs using similarity thresholds of pre-set sequence identity is not risk-free, as evident by the inherent consequence that multiple similar but different species would be grouped and blurred into a single OTU, losing their individual identifications. The assumption that the shared sequence identity border in prokaryotes for the 16S gene should be "near" 97% [5], or to 98.65% [8] or to 98.75% [9], shows how relative this concept is, and is increasingly seen as a rather arbitrary and unstable default choice, whose weaknesses are also evident from the incongruencies that the databases are continuing to reveal [10, 11] which made the 97% value, on which the OTU 97% clustering is referred, just a mere relic of a surpassed convention, being neither a reliable nor an unambiguous set point for the bacterial species discrimination. In this respect, it can be foreseen that with the increase of the sequencing technologies throughput and of the ensuing bioinformatics, the basis for taxonomical assignments will be eventually shifted from the metabarcoded amplicons to the Metagenome-Assembled Genomes (MAGs).

As alternative approaches to the binning into OTUs, some had proposed to require extremely high levels of sequence identity to minimize the loss of diversity when clustering, which however was recognized to potentially mistake the sequencing errors as grounds for false new species attribution [12].

The OTU clustering that was traditionally adopted by the sequencing studies is the reference-based type, which, as opposed to the *de novo* clustering, is a closed-reference operation that draws within the available database of target gene sequences. A sequence that has a high number of discrepancies would prevent any clustering to known subjects and will be discarded by the process. This represents at the same time both a measure against sequencers errors but also a condition that hampers the discovery of genuinely new taxa due to the self-referenced nature of the comparative process. Moreover, if errors exist in the reference database itself,

which does indeed occur at a basal rate, a further level of bias adds up. These caveats made clearer the necessity to develop open-reference clustering, adopting both the principles of the closed-reference method and those of the *de novo* clustering, to avoid loss of the novel taxa. Within this rationale the ASV-based approach was developed to pursue a process which, at least in principle, starts as the opposite of the clustering. Rather than blurring reads into an averaging consensus, the method aims at focusing straight on exact sequences, (which, more realistically means, with the minimum possible reads agglomeration compromise) to determine how many times each variant occurred, combining the result with an error model for the sequencer's run and working out a probability of exactness with a statistical confidence with a p-value for the null-hypothesis that each given sequence were due to a sequencing error. Such choice has also the added benefit that any given target sequence, being an exact variant, is bound to generate the same ASV, which makes the results far more comparable to those from other studies and endowed with a higher resolution for a more precise identification at species level and beyond [13].

Moreover, while OTUs are generally considered to be a suitable protocol to retain sequences that are rare in a sample, they pay the toll of risking a higher rate of picking spurious OTUs [14]. From the ASV side on the contrary the issue has been efficiently addressed since the advent of a variant determination software as DADA2 [6], which is particularly suited for low-abundance reads [15].

In general, the ASV approach has been recognized to be advantageous also for its better performance in the presence of confounding factors as contamination issues where, using community standards at known amounts, it was demonstrated to accurately tell apart sample DNA from contaminant biomass proportions [16].

Also as regards the technical drawback of chimera occurrences, being, in principle, the ASVs assumed to be nearly exact sequences, they do not require to deal with a fuzzy consensus of lumped sequences as the OTUs, because a chimeric ASV, being the 'exact daughter' of two exact parent sequences usually prevalent in abundance within the sample, is easily spotted upon alignment due to its neat junction [6].

Within the scientific community the light on the need to catch the relevance of the newly available resolution offered by ASV was casted rather eagerly by a paper published in ISME Journal with the title: *Exact sequence variants should replace operational taxonomic units in marker-gene data analysis* [17]. The debate has also seen arguments as regards the inference of correct ecological information from sequence data processed with either method. While some authors indicated a substantial reliability of the OTU-based conclusions due to their strong correlation with the ASV [18], others pointed out the biases bound to the use of OTUs in estimating community diversity when compared to the exact sequence variants [19], although in that case the methods used, that introduced other variables in the parallel protocols, endedg up in a condition in which the number of resulting ASV was even lower than the one of the OTUs, which is in itself a contradictory outcome. Other reports found that the two different approaches yielded community compositions that differed between 6.75% and 10.81% between the alternative pipelines [20]. The family rank level was the one where some authors found the most substantial agreement between the two methods, although in general terms ASV would outperform OTUs as regards community diversity [21].

The fact that alpha diversity indices would result lower in OTU-based than in full variants-based calculations might sound simply a consequence inherent in the clustering of OTUs, which inevitably leads to fewer OTUs than ASVs. This is however just the inherently automatic portion of the effects of merging objects to create fewer sets. But the central aspect of the debate is in our opinion to point out that, within this expected trend, the actual loss of diversity extent is concealed, and can pass vastly overlooked until one quantifies it. E.g. if in a set of

1000 sequences there were 100 actually different variants, but they would form only 10 groups sharing within each 97% identity, the resulting number of clustered OTUs would be 10. But if the 1000 sequences had been instead all different (1000 actual variants) but those differences would still occur within the ten 97%-coherent packages, the OTU-based inference could in that case underestimate by tenfold the actual genetic biodiversity of the sample. More explicitly from the numerical point of view, even with reads as short as 100 nucleotides, an OTU clustered at 97% cutoff will have 3 nucleotides free for variation included in the packaged set. The possible combinations accounted by the four possible types of the DNA bases distributed in three positions is $4^3$ = 64 combinations. Therefore, in those reads there is theoretically room for up to a 64-fold underestimation of the hidden diversity when adopting that common extent of clustering, Considering also that actual NGS reads are more realistically twice as long, the resulting possibility for variability in six positions is $4^6$ = 4096 variants.

The ASV approach has nevertheless been critically scrutinized by authors that have remarked how its output is not to be misunderstood as that of truly unique and single sequences. This is because variants generated by the DADA2 step are actually stemming from a process that has also a low but inevitable degree of agglomeration and, strictly speaking, should therefore be regarded simply as a less-clustered type of output, but still on the conceptual continuum of the OTUs themselves [22].

In addition, while the use of ASV has been essentially advocated for bacterial community studies (16S rDNA), when it comes to fungi and the target amplicon is the ITS, different issues as intraspecific, and even intragenomic variability for repeated copies of the spacer, have led to a reversed appreciation of these tools, showing that, for those eukaryotes, OTUs outperform ASV in resolving fungal diversity [23]. Some reports however state contrasting results on this point [21].

Finally, irrespective of which method is used to agglomerate the sequences, downstream of that stands an important warning that applies to metabarcoding studies in general. One needs to keep in mind that the datasets resulting from metagenomics are compositional, meaning that the values (even when in the form of real integer numbers of reads) always represent relative abundances, in which the counts of each taxon are constrained by those of all the others, as in a pie chart in which the sum is always bound to give 100%, calling for crucial data transformation and ensuing bioinformatics which is often still neglected in most reports [24–27].

In this work, dealing with bacteria, we sought to address particularly the aspect of OTUs vs. ASV handling, which can be critical for the applied aspects of environmental impact assessment, or to measure the effects of different agricultural management practices on soil ecology, as well as to compare between natural and anthropized landscapes and their biodiversity gradients.

As elements of novelty in our present analysis, the following apply: (a) differently from prior reports, to avoid pipeline-related proportional alterations, as e.g. DADA2 denoising filtering for ASV vs. straight Mothur-based clustering for the OTUs, we kept constant all aspects of the two parallel processing routes, and we had, as the only differing variable, the presence or absence of the clustering step. The same denoising with DADA2 was thus performed in both cases. Additionally, (b) we analyzed 10 different ecological indicators, most of which are usually ignored in microbial community surveys; (c) we verified the extent of the risk encountered by working with or without primary data transformation to circumvent the dataset compositionality issue, and compared a series of differently bias-sensitive distance metrics; (d) we analyzed also the effect of clustering reads on the correlation between the different ecological indicators; (e) as site of investigation we chose a transect in a single vegetational gradient landscape in order to assess also gamma diversity, leading to the possibility of reporting the effect of the bioinformatics choices on each of the three levels of ecological diversity (alpha, beta and

gamma); (f) we included multivariate approaches as cluster analysis, Principal Coordinate Analysis, as well as a Least Discriminate Analysis Effect Size, that visually showed aspects as the dendrogram tree topology collapse, PCoA biplot significance loss, and differentially featured taxa detection, which occurred rather evidently at 97% OTUs clustering.

The working hypothesis was that the ASV based analysis, being endowed with a higher statistical power conferred by the higher number of data points within each sample, and consequently by an inherently higher resolution of the actual sample diversity, should yield data whose consistency would be progressively affected by the compromising custom of melting the available diversity data into discrete packages of Operational Taxonomic Units.

## Materials and methods

### Samples origin and collection

The location of origin of all samples was the Valle Vecchia demonstrative pilot farm and nature oasis, (province of Venice, Italy), a site under the management of the Veneto Agricoltura, Regional Council Agency for Agriculture Forestry and Fisheries. The 34 samples analyzed covered a vegetational gradient through 17 connected adjacent habitats and their ecotones, from cropped fields through meadows, riparian hedges, forest, floodplain, prairie, sand dunes to seashore, located in the same coastal area of North-Eastern Italy. The transect extended within a 700 m length. The choice of a series of very different (bur spatially connected) habitats was meant to encompass a wide diversity in the resulting database, i.e. a high gamma diversity of the whole area. Such choice was envisaged as a way to maximize the chance of observing diversification, in terms of micro-evolutionary variation within the whole community. A thorough description of the site and of the sampling campaign (performed in September 2016) has been described in our prior report in which the account on the overall taxonomical comparisons and other methods to extract the concealed diversity within data, has been previously presented [28]. For some of the analyses (PCoA, LDA), the 17 habitat types were grouped in the following seven macro-habitats: 'Cropped', 'Prairie', 'Hedges', 'Floodplain', 'Transition', 'Coastal', 'Waters'.

### Sequencing and bioinformatics

Microbial community profiles were determined using the 16S rRNA gene V4 hypervariable region 160 with universal primers (515F/806R). Sequencing (paired ends, with reads length of $2 \times 250$ bp) was performed with an Illumina MiSeq platform at the Ramaciotti Centre for Genomics (Sydney, Australia) generating a total of over 3 million reads. Raw fastq reads were imported into Qiime2-2022.2, and primers (`515F- GTGYCAGCMGCCGCGGTAA` and `806R-GGACTACNVGGGTWTCTAAT`) were trimmed using the Cutadapt Qiime2-2020.2 plugin. Following primer trimming, an average of 95826 reads per sample were obtained and reads were denoised using qiime DADA2 denoising plugin to obtain the ASVs and the counts table. The downstream analysis further was split into two approaches. In the first approach, ASVs obtained from DADA2 denoising, were assigned taxonomic labels using qiime classify-sk-learn and SILVA 138.1 database [29]. The second approach aimed at analyzing sequences based on the operational taxonomic units (OTUs), where the trimmed sequences and denoised using from qiime DADA2 plugin, were followed by OTU clustering with a 97% or a 99% sequence similarity cutoffs using the qiime vsearch plugin. The representative sequences from OTUs were then classified using the classify-sk-learn plugin with the SILVA 138.1 16S SSU database [29]. The raw ASV/OTU counts table and the taxonomic assignments from the qiime environment were exported using the qiime export plugin. The pipeline scheme adopted for

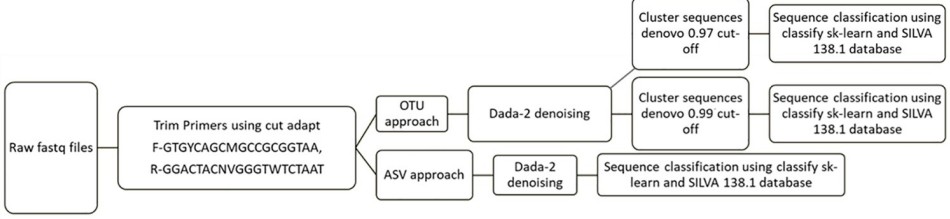

**Fig 1. Bioinformatics processing strategy for the three parallel approaches.**

the different processes is shown in Fig 1 and a detailed script with the commands used are provided in S1 Text.

The pipeline scheme adopted for the different processes is shown in Fig 1.

The ecological indices of Shannon-Wiener H value, Simpson's 1-D, Community evenness (e^H/S), Brillouin, Menhinick, Margalef, Equitability J, Fisher alpha, and Berger-Parker, as well as Centered Log Ratio (CLR) transformation, Neighbor-Joining dendrograms based on Jaccard Distances or on Bray-Curtis Dissimilarity, the Pearson and Spearman Correlation and the Linear Discriminant Analysis (LDA) Effect Size to extract significantly differentially featured taxa, were all computed from the output data matrix using the Past 4.13 software [30]. LDA score values were calculated from the trimmed mean M values (TMM)-transformed data. Alpha-diversity and evenness significance of differences were estimated using the Microbiome Analyst online utility (https://www.microbiomeanalyst.ca/). Principal Coordinate analysis and PERMANOVA for beta diversity assessment testing the following distance metrics: Binomial, Canberra, Clark, Raup, Wave hedges, were performed using the SHAMAN online utility (https://shaman.pasteur.fr).

As regards data transformation, for the analyses performed with the PAST software, data normalization using the Aitchison's centered log-ratio transformation (CLR) was carried out to circumvent compositional dataset constraints. For the PCoA and PERMANOVA analyses, the Weighted Non-Null transformation option of the SHAMAN Suite was selected, while for the LDA executed by the Microbiome Analyst suite, either the CLR (in the PAST Software) or the Trimmed Mean M values (TMM) transformation (in the SHAMAN Suite) were compared.

The sequencing data were deposited in the NCBI Sequence Reads Archive (SRA) under code PRJNA608631.

## Results

### Alpha diversity and community evenness underestimation

The sequencing had yielded 5.886.696 raw reads from which, after the filtering/denoising steps 2.579.452 were retained, 1.315.432 of which could be assigned to taxonomic identities. Comparing the effects of the three procedures, clustering the reads into discrete subpackages, as the OTUs are, caused, as inherently expected, a reduction of their numbers with respect to their full list of single sequence variants. The corresponding datasets in these procedures include therefore fewer and fewer sequences as a function of the chosen percentage of shared homology cutoff, which was regularly observed in the present data. The verified consequence was that of a blended level of identity that, in turn, reduced the resulting diversity since the achievable resolution was deliberately reduced. The indexes that are linked to taxa presence and abundance patterns are therefore, in theory, expected to be susceptible to these changes, which

**Table 1. Alpha diversity result loss caused by each of the two clustering choices in comparison to full ASV data analysis.** The percent values with respect to those stemming from the taxa counts of the ASV table are shown. Values are the means from 34 samples collected across the different habitats of the habitat type gradient.

|  | ASV | OTU 99 | OTU 97 |
|---|---|---|---|
| n. taxa | 100% | 95,7% | 83,6% |
| Simpson_1-D | 100% | 99,9% | 99,5% |
| Shannon_H | 100% | 98,0% | 92,3% |
| Evenness_e^H/S | 100% | 93,7% | 76,3% |
| Brillouin | 100% | 98,1% | 92,4% |
| Menhinick | 100% | 95,7% | 83,6% |
| Margalef | 100% | 95,6% | 83,5% |
| Equitability_J | 100% | 98,8% | 95,0% |
| Fisher_alpha | 100% | 94,9% | 80,5% |
| Berger-Parker | 100% | 119,3% | 183,3% |

was confirmed by our analysis. Table 1 reports the effects of the progressive clustering through 99% and 97%, which is, as mentioned, the commonly used OTU standard. Data are stemming from the total of 13073 ASV distributed across 34 sampling points covering an ecological gradient of habitats from cropped fields to the seaside and consequently warranting a wide range of diversity variation. Data are expressed as percentages of the values scored by the ASV dataset.

Besides the straight reduction of over 16% of the number of different taxa, the behaviour of the ecological indexes appears very uneven. While the Simpson 1-D index [31] was only minimally affected (being based on the complement of dominance, and essentially concerning the probability that two taxa taken randomly from the community, represent the same kind), conversely, the equally popular Shannon-Wiener index [32] is barely reaching the 92% of its corresponding value when the full variants are considered. The latter index, in comparison to the former, is recognized as the one more closely reflecting the community structure as it takes into account both the number of taxa and that of individuals. The Shannon-Wiener index however starts from the theoretical assumption that individuals are randomly sampled from an 'infinite' population and that all taxa would have to be featured in the sample An inborn source of bias in such index arises therefore from the failure to possibly have all taxa in a sample, and this error increases progressively as the proportion of species discovered in the sample declines [33, 34].

Even more dramatic is the effect on another major indicator used in community ecology, which is the Evenness value (e^H/S, [35]), dropping to just 76.3%. Other indexes are all variably affected; the Brillouin measure [36] takes into account the number of observations and the number of individuals belonging to the most abundant taxon and the total number of taxa, and results underestimated by a factor similar to that affecting the Shannon values. A further higher discrepancy results also for the Menhinick richness index, which is the number of taxa divided by the square root of the individuals [37] and Margalef's richness, computing the number of taxa -1 divided by the natural logarithm of the number of individuals [38].

The Equitability J parameter is instead the Shannon diversity divided by the logarithm of the number of taxa, and accounts for the partitioning level by which individuals are spread among the species present [32], while the Fisher alpha index is tied to a slope constant of the distribution [39].

Besides the underestimation of richness-proportional measures, it is relevant to remark that the Berger Parker indicator, based the number of individuals in the dominant taxon relative to

the total number of individuals in the community [40], which is the only one that regards dominance, and has therefore the opposite meaning in ecological terms (the lower the better), displays an extremely inflated effect that boosts it to nearly a double value (183%) when using the 95% clustered OTUs.

The data shown in Table 1 are moreover the average ones as the situation can be far more extreme depending on the community structure of a given habitat. In this respect the minima observed can be as low as 64%, as e.g. for the case of community evenness, which, conversely, is known to suffer of an overestimation for samples in which the total number of different taxa is particularly low [35]. The full dataset of ecological indexes is available as Supporting Information (S1 Dataset Ecological Indexes).

## Effects of clustering on ecological indexes correlations

A further aspect on the alpha diversity context emerged by the pairwise correlation analyses among the different ecological indexes obtained from each of the three progressively clustered taxa tables. The sequences abundance and the number of resulting taxa were also included in the crossed comparisons, whose results are shown in Fig 2. As a premise it needs to be recalled that the statistical power of an analysis is a function of the sample size and that therefore using

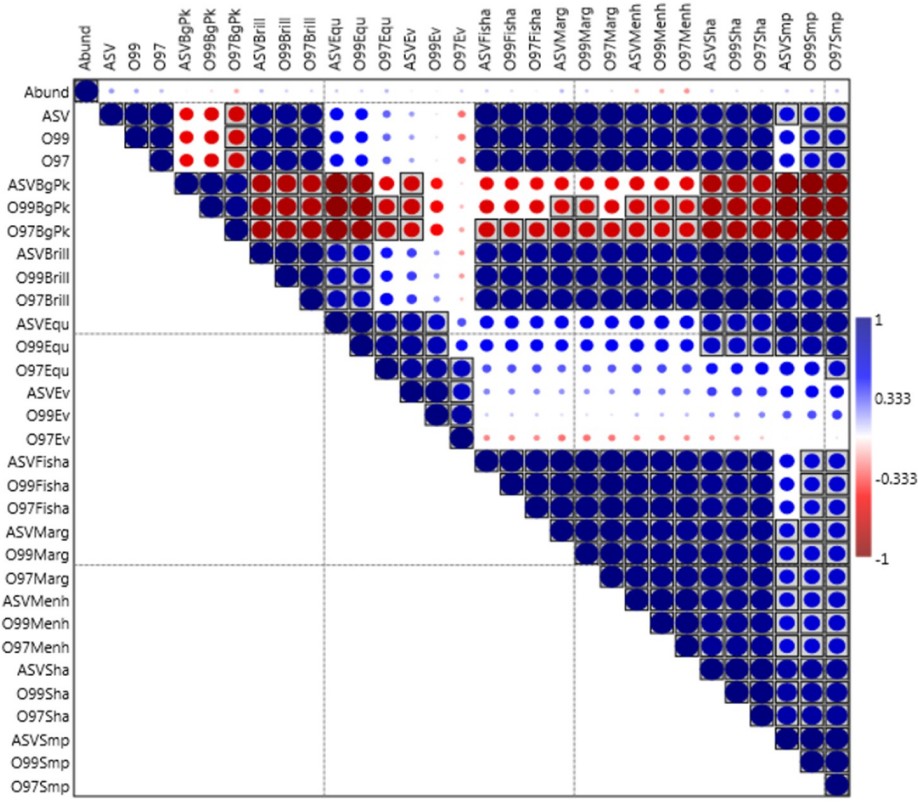

**Fig 2. Correlation matrix (Pearson coefficient with Bonferroni-corrected p values) of the pairwise comparisons across numbers of taxa and ecological indexes of the three sequence clustering approaches.** Boxed cells with grey background indicate significant differences for p<0.05. Abund: sequence reads abundance; ASV, O99, O97: number of different taxa resulting from the full ASV analysis or from the OTU clustering at 99% or 97% shared homology, respectively. These three prefixes apply for the remaining correlation indexes' abbreviations, whose suffixes indicate the following: BgPk: Berger-Parker Dominance; Brill: Brillouin Diversity Index; Equ: Equitability J; Ev: Community Evenness e^H/S; Fisha: Fisher alpha Diversity Index; Marg: Margalef Richness Index; Menh: Menhinick richness index; Sha: Shannon-Wiener H Diversity Index; Smp: Simpson 1-D Diversity Index.

ASV which are always more numerous of the OTUs, is bound to guarantee a higher accuracy and a consequently higher possibility of finding true correlations when those exist.

The analysis unraveled differences in their significance stability as a function of clustering and also unexpected inversions of sign in the correlative direction, i.e., positive correlation turning negative as in the case of community evenness.

In first instance a weak relationship between sequencing depth (reads abundance) and all indexes' outputs appears, displaying in most cases a positive sign with the expected exception of Berger-Parker dominance and the less intuitive exception of the Menhinick richness, that is linked to the denominator position of the square root of the individuals.

As regards the positive correlations (blue spots) it can be noticed that in many instances the strength of the correlation is weakened along with the clustering intensity and in some cases a significance that was recorded using the ASV data is lost when OTUs are the clustered units. It is the case of Equitability and of both Shannon and Simpson indices, and even of the latter index itself in its relationship between the ASV data with either the number of OTUs at 99% or OTUs at 97%, which, by definition, ought to be strongly correlated with it.

As regards differences in the correlation direction, an opposite sign in pairwise comparisons among these indexes (red spots) is expected only between dominance indicators (i.e. Berger-Parker) and all the others which are instead representing diversity or evenness. The data comply, but at the same time show also an equally opposite phenomenon in terms of effects of the clustering: the significance tends in this case attributed to the comparisons involving the progressively most clustered cases (OTU 97%) and excluded in crosses involving the indexes that were calculated with more available data (ASV) and that are therefore endowed with an inherently higher statistical power. The fact that those significance-scoring correlations would pop up just in crosses involving the Berger-Parked Dominance is tied up to the above signalled bias of inflated estimation of the index itself, as seen in Table 1, where an off-scale value of 183% was occurring for the 97% OTUs as a fraction of the same index when calculated from the more data-rich ASV dataset.

But the most striking incongruency observed from the correlation table is the behaviour of the Evenness parameter, as that is the only one that even showed a sign inversion along with the clustering percentage cutoff reduction displaying the whole gradual change from an extreme to the other. That is well visible (Fig 2) in the nine crossed comparisons between the three evenness values (ASVEv, O99Ev, O97Ev) and the three taxa numbers resulting from each set (ASV, O99, O97), in which a positive correlation (blue) occurs with ASVEv, an absence of correlation (blank) with the O99Ev and a negative correlation (red) with the O97Ev. Analogous inversions are visible across the same Evenness parameter and the indexes of Brillouin, Fisher alpha, Margalef, Menhininck and Shannon, for all of which the correlation with the evenness calculated from the OTU 97% clustered units paradoxically assumes a negative correlation outcome unlike the cases of its ASV and OTU 99% counterparts.

The corresponding non-parametric version of the same correlation analysis was carried out using the Spearman Coefficient (S1 Fig) which yielded correspondingly analogous results.

## Beta diversity: Effects of clustering on apparent distances in multivariate ordination

To analyze the between-samples difference, keeping in consideration the caveats on possible biases due to dataset compositionality [24–27] we tested two approaches, the first adopting the Centerd Log Ratio (CLR) transformation and the Jaccard distance measure, and the second relying on the pairwise Bray-Curtis distances from primary untransformed data. Both procedures were used for each of the three sequence processing cases to run a multivariate cluster

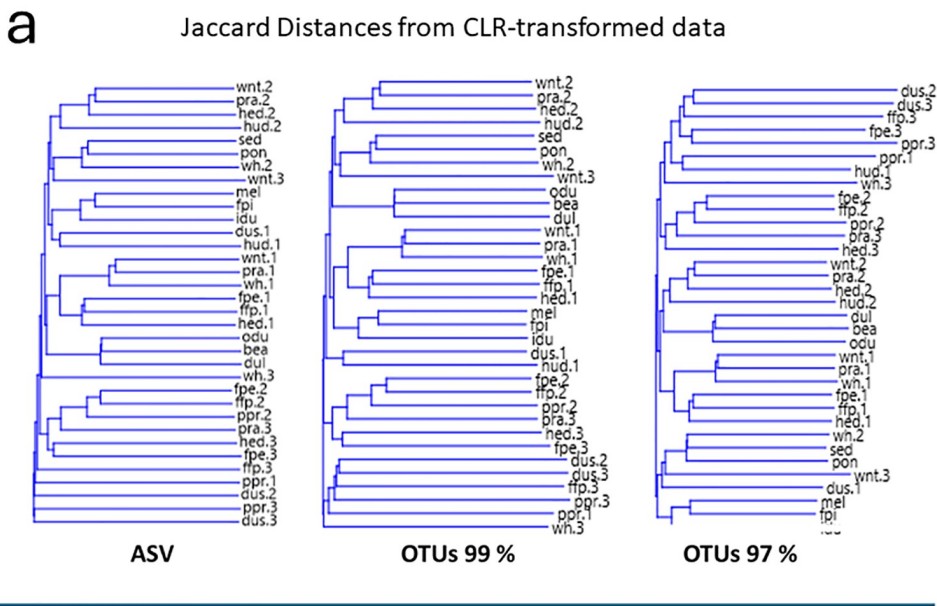

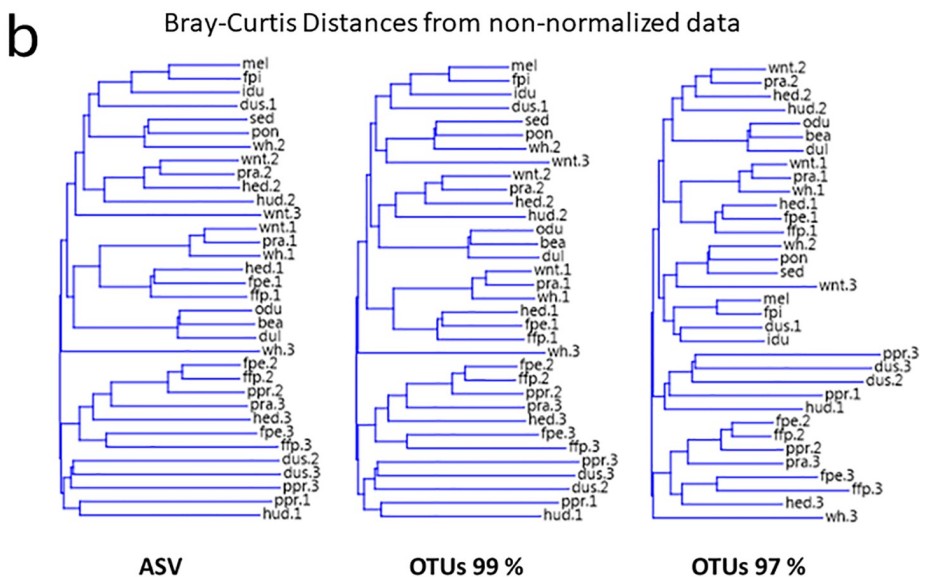

**Fig 3. Neighbor Joining dendrograms obtained by the cluster analysis of each of the three different data matrix tables.** a) Jaccard distances resulting from the Centered Log Ratio (CLR) transformation of the data to circumvent compositional constraints; b) Bray-Curtis dissimilarity-based dendrograms from non-normalized data. In either case the holding consistency of the tree topology at OTU 99% clustering is visible, along with its evident loss when assembling sequences in the 97% clustered packages. Samples nomenclature is drawn from [28]. Habitat acronyms: fpe: forest poplar elms; fpi: forest pines; hed: hedges; hud: humid dune; idu: inner dunes; mel: meadow on levee; odu: outer dunes; pon: pond; ppr: pines prairie; pra: prairie; sed: sea sediment; wh.: wheat; wnt: wheat no tillage.

analysis ordination with the Neighbor Joining criterion to produce the resulting dendrograms. As can be seen by inspecting the resulting phenons (Fig 3) the grouping of samples, when comparing the ASV-based ordination with the one obtained by the minimal clustering (99%), is substantially consistent, while the adoption of the standardly used 97% OTU clustering, results instead in a number of changes in the samples relative positions, in an alteration of the horizontal distances, and in a marked overall change of shape of such distance/similarity-based

**Table 2. PERMANOVA p values associated to the beta-diversity Principal Coordinate Analysis (PCoA) results using each of the three dataset matrixes.** Samples have been grouped into seven ecologically coherent sets across the land-to-sea transect (Cropped, Prairie, Hedges, Floodplain Transition, Coastal, Waters) among which the community diversity was assessed. P values significant for p<0.05 are in bold and marked with an asterisk (*).

| Distance metrics | ASV | OTU 99 | OTU 97 |
|---|---|---|---|
| Binomial | **0.033*** | **0.039*** | 0.076 |
| Canberra | **0.028*** | **0.029*** | **0.038*** |
| Clark | **0.038*** | **0.045*** | 0.070 |
| Raup | **0.019*** | 0.086 | 0.540 |
| Wave hedges | **0.025*** | **0.025*** | 0.051 |

The table allows to appreciate that only at the ASV-grade output the significance is maintained for all five tested distance metrics used to ordinate the data matrixes, while the commonly used OTU-97 reads assembly scores a p value < 0.05 only for the Canberra distance measurement.

dendrogram, underlining once again the effects of the clustering choice. Effects were noticeable both for the untransformed and with the CLR-transformed data.

A further approach to visualize effects on beta diversity was to construct Principal Coordinate Analysis (PCoA) ordination biplots and testing five different distance metrics (Binomial, Canberra, Clark, Raup, Wave hedges) with each of the three dataset matrixes. In addition to this, a Permutational ANOVA (PERMANOVA) analysis was run on the ordination data to inspect the significance of the differences. Results of the latter are shown in Table 2, in which the beta-diversity was calculated among seven ecologically coherent sets across the land-to-sea transect (Cropped, Prairie, Hedges, Floodplain Transition, Coastal, Waters) within which the 34 samples were grouped. As regards the PCoA, the three biplots obtained by the Raup distance, that was visually the most informative example, are shown in Fig 4.

It can be appreciated that the degree of resolution separating the sets of each macrohabitat in the Principal Coordinate Analysis is progressively shrinking along with the increasingly clustered ordination and that the significance is correspondingly fading.

Subsequently we calculated the Linear Discriminant Analysis Effect Size evidencing the significantly (p<0.05) differentially featured taxa from each of the three dataset matrixes, their LDA scores and the up or-down representation of each taxon within the seven macrohabitats. The results are shown in Fig 5.

It can be noticed that the number of differentially featured taxa scoring significant differences is progressively decreasing when moving from ASV to OTU-99 to OTU-97 passing respectively from 7 to 6 to 3.

## Gamma diversity drops upon reads clustering

Having chosen for the present survey a well-defined area encompassing a continuous transect of several adjacent habitats and their respective transitional ecotones, we can take into account the full range of diversity arisen from the metabarcoding through the Valle Vecchia Oasis i.e. the resulting Gamma diversity of the whole site. In this respect the total number of non-redundant different taxa counted among the whole series of samples amounted to the values reported in Table 3, which lists also the relative proportions of the two OTU-based analyses and the consequent taxa loss in when compared to the ASV-based results.

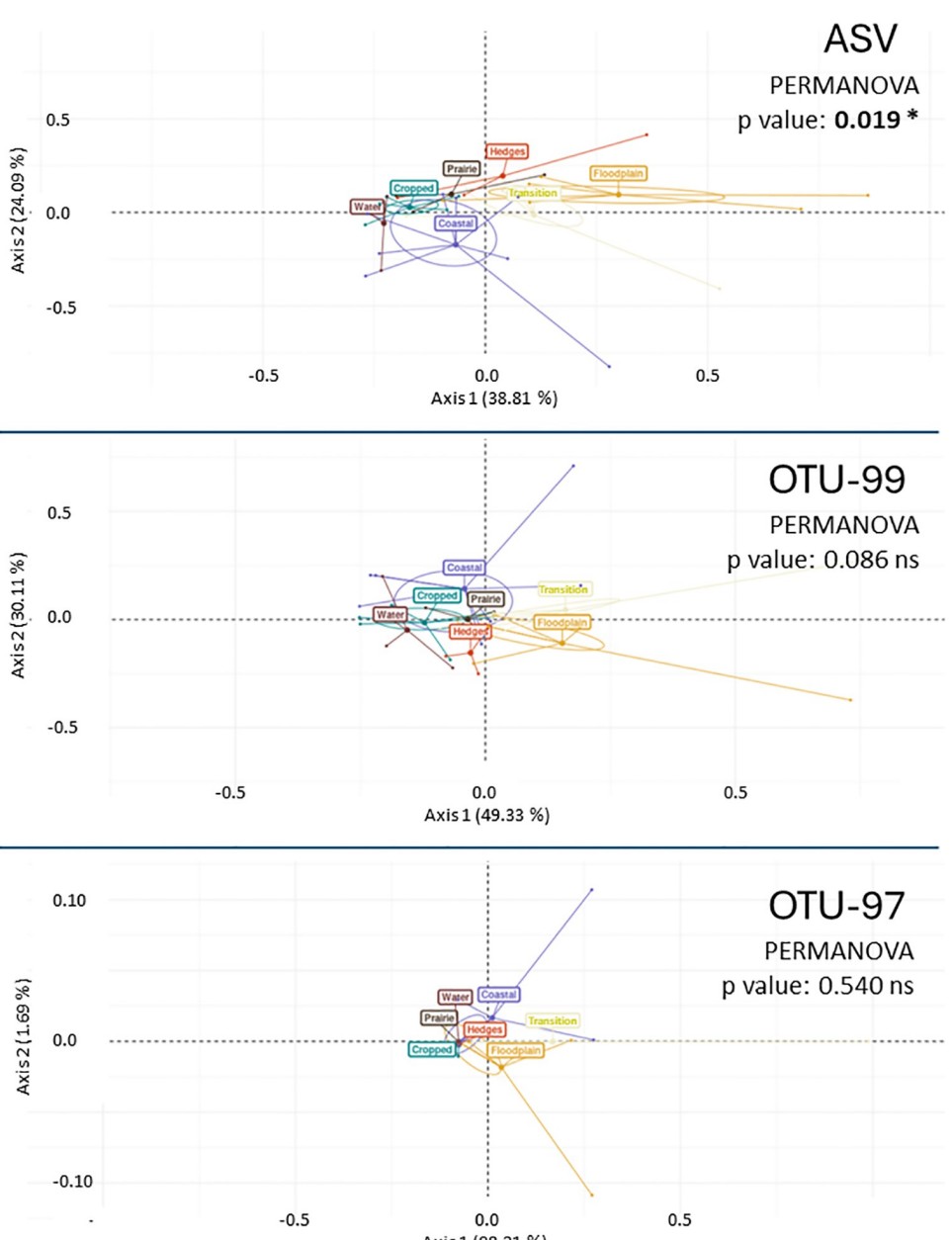

**Fig 4. Principal Coordinate Analysis (PCoA) ordination biplots based on the Raup distance metrics with each of the three dataset matrixes.** The beta-diversity is calculated among seven ecologically coherent sets across the land-to-sea transect (Cropped, Prairie, Hedges, Floodplain Transition, Coastal, Waters) within which the 34 samples were grouped.

## Discussion

The analysis performed has evidenced several critical differences that endorse the use of exact Amplicon Sequence Variants, although not immune from a limited degree of reads agglomeration [22], as preferable with respect to Operational Taxonomical Units, at least for a prokaryotic dataset. The clustering mechanism that underlies the latter, besides the consequent reduction of the diversity output which is inevitably assumed by the procedure in itself, shows

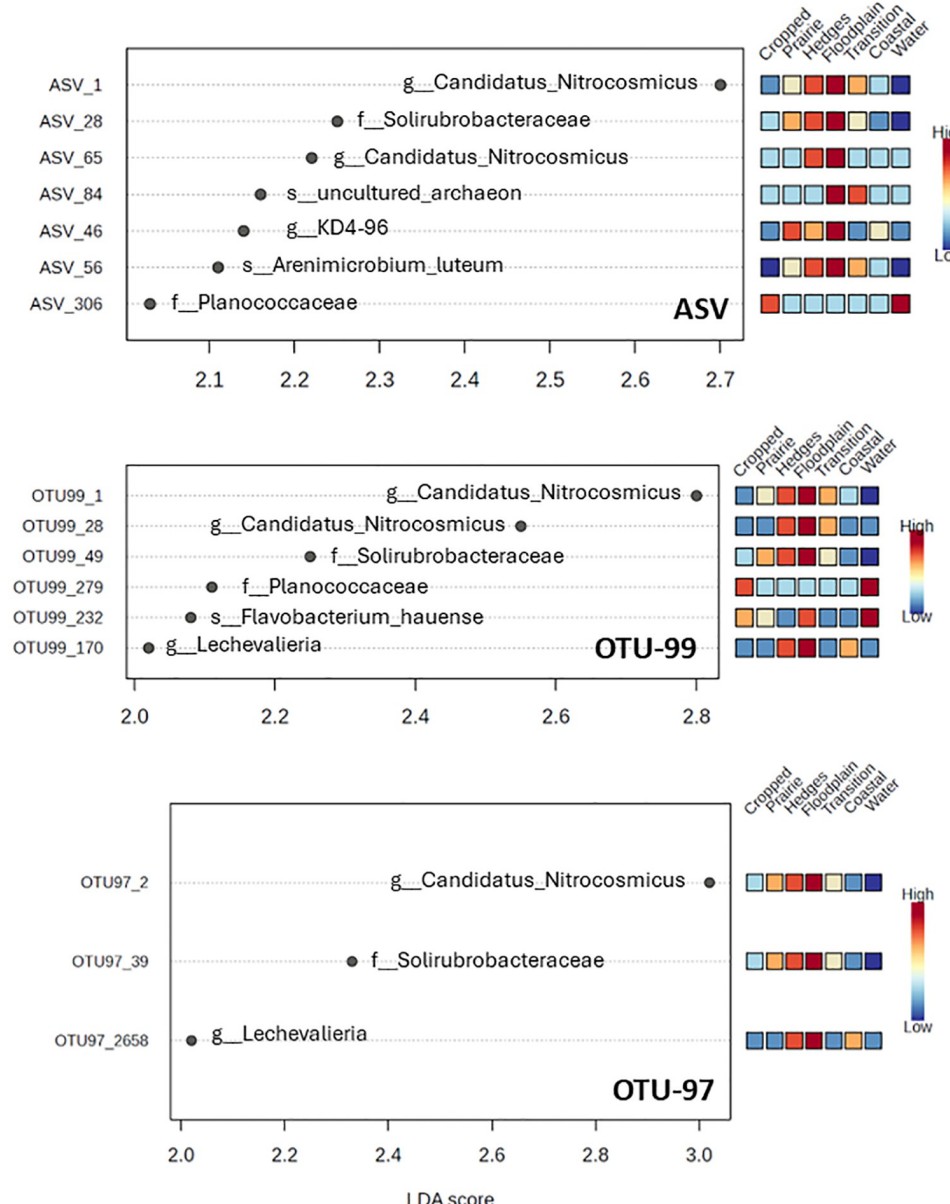

**Fig 5. Linear Discriminant Analysis Effect Size evidencing the significantly (p<0.05) differentially featured taxa from each of the three dataset matrixes.** Top: ASV, Middle: OTU_99, bottom: OTU_97. The relative occurrence of each taxon across each of the seven ecological habitat subsets encountered stepwise along the transect is reported on the right side. The LDA score values are calculated from the trimmed mean M values (TMM)-transformed data.

limits in the estimation and comparability of ecological diversity indexes which should instead constitute the actual goals in metabarcoded microbiome comparisons in applied studies. The underestimation of richness-based parameters when compared to the corresponding values stemming from ASV, and the inflation of the ones based on dominance, as well as the loss or reduction of significance and correlations among indexes and data outputs, suggest to apply care when dealing with the clustered packages of data which are the basic units in the OTU-assembling procedure. The inversion of correlation coefficients direction observed for

**Table 3. Gamma diversity of the whole site.** The total number of non-redundant different taxa counted among the whole series of samples is reported and the corresponding comparative levels for the OTU-clustered datasets in comparison to the ASV values.

| Taxa | ASV | OTU 99 | OTU 97 |
|---|---|---|---|
| Total number | 13073 | 10515 | 7275 |
| Percent of ASV value | 100% | 80.4% | 55.6% |
| Richness (loss) | 0% | -19.6% | -44.3% |

community evenness across the two sequence analysis methods is in this respect particularly emblematic of the existing discrepancy and of the ensuing interpretational risks.

In our analyses we also considered the risks stemming from the general constraint of dataset compositionality that concerns all metagenomics. To assess the extent of those we investigated whether major differences would arise by using straight primary data matrixes as opposed to performing appropriate transformations to reduce that bias. In particular, for the cluster analyses comparison we compared the effect of data transformation using the Aitchison's centered log-ratio transformation (CLR) which was recommended to avoid as much as possible the consequences from dataset compositionality [24]. Furthermore, instead of the Bray-Curtis distances, the cluster analysis from the CLR-transformed data was carried out using the Jaccard distance metrics, which is less prone to that effect since it does not use abundance data but only presence/absence. Thus, we compared the effects on Neighbor Joining tree topology consistency with those seen upon calculating Bray-Curtis Dissimilarities on primary data. In both cases the degeneracy of tree stability culminating at 97% clustering was nevertheless observed.

Our data are in essence very consistent with the arguments in favor of moving the field towards the choice of ASV (Callahan et al., 2017, Caruso et al., 2019), at least for 16S-targeting investigations, due to the more precise identification of bacteria within communities as well as in providing a crisper picture of the actual environmental diversity within each sample. On the contrary, an OTU, embodying a cluster of multiple reads could contain both real sequences or errors lumped together into an arbitrarily assembled unit, introduces a process bias that can be avoided by the handling of the separate single variants.

The trade-off between the computationally easier generation of OTUs appears surpassed by the issue of reference-related biases, and in particular when the comparison involves the traditional closed-reference OTUs which are prone to miss novel sequences. While that could be acceptable in projects whose target is an already well-defined database of records, as, e.g., in human microbiome surveys [41] where the expected taxa are nowadays at the most within acquired knowledge, the situation is very different for the explorative environmental microbiology, where soils, oceans and most yet poorly-known habitats, contain arrays of uncultured and unknown taxa that we cannot afford to miss due to closed self-referencing annotation methods. Alternatives as the complex and machine memory-intensive *de novo* OTUs clustering or mixed open referenced and closed references OTU-based approaches appear less advantageous that the ASV approach, both for the implementation of databases with new data and in terms of accuracy, since the choice of accepting only high-confidence exact variants can be exerted.

Furthermore, from the computational point of view, while OTU generation may appear an easier task in comparison to ASV, the latter approach allows to update the ASV/taxa table with new incoming data without the need to re-analyze the full data all over since it is based on exact sequences. Whereas in the case of OTUs, the clustering needs to be performed again to add new data.

Undoubtedly OTUs have contributed enormously to the build-up of our microbiome knowledge and for some scopes as the gut microbiomes of well-characterized species as humans, livestock animals, or laboratory model species, they will keep being preferable in terms of big data handling from population wide studies and other designs. For ITS, in surveys seeking fungal diversity arguments in favor of the performance of OTUs are still prevalent [23] when compared to the opposite [21]. But on the other hand, the maturity reached by bioinformatics applications as the mentioned DADA2, or the improved denoising procedures [15] and finishing tools as Deblur [42] along with error corrective ones as UNOISE [43] have paved the way for an increasingly trustworthy adoption of the amplicon sequence variants in bacterial metabarcoding.

In conclusion, we have put in evidence from different standpoints the advantages that support the use of 16S amplicon variants. As advancements and novel elements over previous reports we have here chosen to adopt fully parallel protocols for the two approaches on the same dataset, by applying the same denoising operations to the original FASTq outputs and using the clustering at the two levels of similarity as the sole variable. Moreover we explored the behaviour of ten independent ecological indexes based on rather different formulas, of three types of data transformation, and of seven different distance metrics to point out the variable extent of bias that occurred with each, to which we added a multiple pairwise correlation analysis, including the relationships among the different indexes themselves, which, to our knowledge, had not been precedented in literature. The visual observation of the consequences of clustering via tree topology restructuring, PCoA consistency loss, p value significance drop, differentially represented taxa detection decrease, culminating at the 97% threshold, in spite of its wide use as routine clustering cutoff, are further elements of novelty. Furthermore, the chosen site offered an extremely variable but spatially-continuous gradient of habitats across a vegetational transect from agricultural crops through forest and seashore, all within the same iso-climatic and iso-geological setting. The elements that arose concur against the null hypothesis of a possible non-significance of differences between results based on OTUs vs. those obtainable by ASV and support the preference of the latter in this type of environmental studies.

## Supporting information

**S1 Fig. Correlation matrix (Spearman Rank Sum Coefficient with Bonferroni-corrected p values) of the pairwise comparisons across numbers of taxa and ecological indexes of the three sequence clustering approaches.** Boxed cells with grey background indicate significant differences for p<0.05. Abund: sequence reads abundance; ASV, O99, O97: number of different taxa resulting from the full ASV analysis or from the OTU clustering at 99% or 97% shared homology, respectively. These three prefixes apply for the remaining correlation indexes abbreviations whose suffixes indicate the following: BgPk: Berger-Parker Dominance; Brill: Brillouin Diversity Index; Equ: Equitability J; Ev: Community Evenness e^H/S; Fisha: Fisher alpha Diversity Index; Marg: Margalef Richness Index; Menh: Menhinick richness index; Sha: Shannon-Wiener H Diversity Index; Smp: Simpson 1-D Diversity Index.
(DOCX)

**S1 Dataset. Ecological indexes.** The alpha diversity and evenness indexes computed from the ASV and OUT tables at the different clustering levels for each of the samples are shown.
(XLSX)

**S1 Text. Bioinformatics pipeline scheme and command scripts.**
(DOCX)

## Author Contributions

**Conceptualization:** Andrea Squartini.

**Data curation:** Andrea Fasolo, Saptarathi Deb.

**Formal analysis:** Andrea Fasolo, Saptarathi Deb, Andrea Squartini.

**Funding acquisition:** Piergiorgio Stevanato, Giuseppe Concheri.

**Investigation:** Andrea Fasolo, Saptarathi Deb, Andrea Squartini.

**Methodology:** Andrea Squartini.

**Project administration:** Piergiorgio Stevanato, Giuseppe Concheri.

**Resources:** Piergiorgio Stevanato, Giuseppe Concheri.

**Supervision:** Andrea Squartini.

**Validation:** Andrea Squartini.

**Writing – original draft:** Andrea Squartini.

**Writing – review & editing:** Andrea Fasolo, Saptarathi Deb, Piergiorgio Stevanato, Giuseppe Concheri, Andrea Squartini.

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
