## [Decision Letter · Decision Letter 0]

24 Apr 2024

PONE-D-23-43692ASV vs OTUs clustering: effects on alpha, beta and gamma diversities in microbiome metabarcoding studiesPLOS ONE

Dear Dr. Squartini,

Thank you for submitting your manuscript to PLOS ONE. After careful consideration, we feel that it has merit but does not fully meet PLOS ONE’s publication criteria as it currently stands. Therefore, we invite you to submit a revised version of the manuscript that addresses the points raised during the review process.

We look forward to receiving your revised manuscript.

Kind regards,

Federico Vita

Academic Editor

PLOS ONE

Journal Requirements:

3. Thank you for stating the following financial disclosure: "Co-author G.C. was the recipient of a national grant from the University of Padova: "Progetto di Ateneo PRAT CPDA154841/15""

4. Thank you for stating the following in the Acknowledgments Section of your manuscript: "This project was funded by the University of Padova in the framework of the ‘Progetto di Ateneo PRAT CPDA154841/15’."

Please remove any funding-related text from the manuscript and let us know how you would like to update your Funding Statement. Currently, your Funding Statement reads as follows: "Co-author G.C. was the recipient of a national grant from the University of Padova: "Progetto di Ateneo PRAT CPDA154841/15""

Additional Editor Comments:

Dear authors,

the manuscript has been revised by two experts in the bioinformatic research field. Both the reviewers raised a series of concerns about your manuscript, which include;

- lack of details in the methods section, including missing details in the sample description and description of diversity measures

- incomplete reference section, which needs to be improved by following the reviewers' comments

- criticism about the use of a specific method (Bray Curtis) for distance matrix calculation

- poorly described results (e.g., gamma diversity ones)

In light of the reviewers' comments found at the bottom of this letter and my own assessment, I recommend major revision.

Reviewers' comments:

Reviewer's Responses to Questions

**Comments to the Author**

1. Is the manuscript technically sound, and do the data support the conclusions?

Reviewer #1: Partly

Reviewer #2: No

2. Has the statistical analysis been performed appropriately and rigorously? 

Reviewer #1: No

Reviewer #2: No

3. Have the authors made all data underlying the findings in their manuscript fully available?

Reviewer #1: No

Reviewer #2: Yes

4. Is the manuscript presented in an intelligible fashion and written in standard English?

Reviewer #1: Yes

Reviewer #2: Yes

5. Review Comments to the Author

Reviewer #1: This is the first round of revision for the manuscript PONE-D-23-43692, entitled "ASV vs OTUs clustering: effects on alpha, beta and gamma diversities in microbiome metabarcoding studies", by Fasolo A and colleagues. In this paper the Authors try to bring their contribution to the ongoing discussion regarding the opportunity or not to work with ASV instead of OTUS in metabarcoding studies, showing a work focusing on the bacterial fraction of coastal environments.

The design of the experiments seems sound, and the manuscript is supported by adequate illustrations, though not very attractive, and supporting materials. Nonetheless, the work shows quite serious weaknesses, which need to be addressed before the manuscript can be accepted.

1. In light of the fact that, as mentioned, there is a fervent debate on whether or not to use ASV in numerical ecology studies, the work necessarily needs to be better framed. On the Fungal side, therefore in relation to the ITS marker, there are milestone works that support the validity of the use of OTUs (Kauserud 2023, Tedersoo et al. 2022), due to intraspecific variability in the ITS region, and sometimes intragenomic variability. The Authors must state much more clearly that their work refers to studies on bacterial communities, and introduce/discuss their arguments in more detail by referring to the opposite findings shown by the two works cited, both in the introduction and in the discussion.

[ https://www.sciencedirect.com/science/article/pii/S175450482300051X?via%3Dihub ]

[ https://onlinelibrary.wiley.com/doi/10.1111/mec.16460 ]

2. Line 68. I agree with this statement, but the problem is also how informative the molecular reference target is. It is also true that if we extend the discussion to other microorganisms we must take into account that when reasoning in terms of identity it sometimes happens that the "intra" variability exceeds the "inter" variability and only a phylogenetic approach can help to reconsider the placement.

3. The methods section is rather superficial, in particular regarding the bioinformatic analyses that follow the taxonomic assignment: there is no mention of the tools used for the calculation of the indices and other statistics, whether included in Qiime or not . Nowadays it is mandatory to provide all the information so that a certain analysis is repeatable; in this regard, I also suggest sharing the analytical workflow and the code used on github or other code sharing platform.

Furthermore, the missing information includes any method of normalization of the data or, alternatively, the justification for why it was not carried out (e.g. rarefaction or other methods). This is particularly important for understanding the indices and their ecological value/significance.

4. In materials and methods, the description of the samples needs to be improved: as a reader I should not be forced to search previous work to obtain information about the nature of the samples. I have no idea, for example, what the fpi sample is. Looking at the S1 table, it seems that some samples comes in replicate, some others not: why?

There is no clearly stated sampling date (2016?). Raw sequencing data values are missing, as well as the final size of the samples after reads cleaning.

Fig. 1 must be modified to report both clustering values of the OTUs.

5. Isn't the fact that alpha diversity indices drop simply a consequence inherent in the clustering of OTUs, which inevitably leads to fewer OTUs than ASVs and therefore to a reduction in index values?

6. Is the phylogenetic tree in Fig.3 the best way to represent the effects on beta diversity? The Authors could delve deeper and explain better how and how much the ordination (b-diversity) changes, and better explain how the cluster analysis varies post-ordination (ASV, 97 and 99%), e.g. Which of the 3 methods generates clusters more consistent with multivariate orderings?

7. The introduction at line 335 of the 'Valle Vecchia Oasis' is misleading, and should be done earlier.

8. I believe that reporting the results of the gamma-diversity only to the entire area and the absence of a graphical representation is limiting, and makes this part of the results unattractive.

9. The results are discussed as if they were generally applicable in metabarcoding studies, favoring the use of ASV more, but this has been shown not to apply for fungi and when using ITS as a target. The entire discussion must be reviewed with the awareness of having obtained contrasting results, highlighting the genomic causes that may have lead to these different results for Prokaryoti and Eukaryoti.

For these reasons, I recommend major revision.

Reviewer #2: Fasolo et. al assessed the effect of different clustering methods (OTU vs ASV) on the diversity measures for microbiome data. The authors applied an amplicon-based metagenomics approach on samples collected from different habitats along a transect of 700 m and assessed the different clustering methods and different diversity measures on them. Among the main findings that the authors portray in the manuscript is the underestimation of the ecological indicators using reference based OTU clustering, therefore favoring the use of ASVs over OTU clustering for analyzing microbiome data. Although the idea of the manuscript is not totally novel and some similar papers have been published in recent years as Chiarello et al. 2022, Jeske and Gallert 2022 and Joos et al. 2020. I like the idea to increase the knowledge on this topic and evaluated this kind of methods in many environments as possible. However, I have some comments and some concerns in the way that data was analyzed that I will proceed to describe next:

- The authors describe in lines 153-155 that in their previous publication there is a detailed description of how diversity was estimated, but I would expect at least a brief description of software, packages and the utilized parameters because not everyone might be interested in your other paper. Moreover, the previous paper of the authors does not contain any description of the diversity measures portrayed in the current manuscript.

- In lines 250-296 it is described a correlation analysis among the diversity indexes in relation to the different clustering methods. I find this analysis interesting, and I understand in principle what its goal is, it is not clear whether these correlations are between absolute count values and different output for each one of the indexes, which makes confusing to me its general purpose.

- In lines 314-321 for the beta diversity the authors created a distance matrix using Bray-Curtis. However, this approach for assessing microbiome data is not correct due to the compositionality of microbiome data, which requires proper methods to draw conclusion at the end. This has been described in several papers as Gloor et. al 2017, Quinn et. al 2018, Susin et. al 2020 and Tsilimigras et. al 2016. Therefore, I will encourage the authors to redo that part of your analyses in the proper way as an improvement to your paper. Moreover, I will suggest too, to perform some PERMANOVA test to see if the clustering method also affects the ability of this type of statistical test to address differences in different groups.

- In lines 333-340 it is described the gamma diversity results, nonetheless, these results are poorly described and not portrayed in figure or table in the manuscript.

- Lines 162 and 172 the name of the software DADA2 is written in 2 different ways

References to check:

Chiarello M, McCauley M, Villéger S, Jackson CR. Ranking the biases: The choice of OTUs vs. ASVs in 16S rRNA amplicon data analysis has stronger effects on diversity measures than rarefaction and OTU identity threshold. PLoS ONE 17,2 (2022): e0264443. https://doi.org/10.1371/journal.pone.0264443

Jeske, J.T.; Gallert, C. Microbiome Analysis via OTU and ASV-Based Pipelines—A Comparative Interpretation of Ecological Data in WWTP Systems. Bioengineering, 9,146 (2022). https://doi.org/10.3390/ bioengineering9040146

Joos, L., Beirinckx, S., Haegeman, A. et al. Daring to be differential: metabarcoding analysis of soil and plant-related microbial communities using amplicon sequence variants and operational taxonomical units. BMC Genomics 21, 733 (2020). https://doi.org/10.1186/s12864-020-07126-4

Gloor, Gregory B., Jean M. Macklaim, Vera Pawlowsky-Glahn, and Juan J. Egozcue. 2017. “Microbiome Datasets Are Compositional: And This Is Not Optional.” Frontiers in Microbiology 8(NOV):1–6. doi: 10.3389/fmicb.2017.02224.

Quinn, Thomas P., Ionas Erb, Mark F. Richardson, and Tamsyn M. Crowley. 2018. “Understanding Sequencing Data as Compositions: An Outlook and Review.” Bioinformatics 34(16):2870–78. doi: 10.1093/bioinformatics/bty175.

Susin, Antoni, Yiwen Wang, Kim-Anh Lê Cao, and M. Luz Calle. 2020. “Variable Selection in Microbiome Compositional Data Analysis.” NAR Genomics and Bioinformatics 2(2):5–7. doi: 10.1093/nargab/lqaa029.

Tsilimigras, Matthew C. B., and Anthony A. Fodor. 2016. “Compositional Data Analysis of the Microbiome: Fundamentals, Tools, and Challenges.” Annals of Epidemiology 26(5):330–35. doi: 10.1016/j.annepidem.2016.03.002.

6. PLOS authors have the option to publish the peer review history of their article (what does this mean?). If published, this will include your full peer review and any attached files.

Reviewer #1: No

Reviewer #2: No

---

## [Author Response · Author response to Decision Letter 0]

7 Jun 2024

FOR AN EASIER READING REFER TO UPLOADED Response to Reviewers WORD DOCX (IN COLOR)

PONE-D-23-43692

Review Comments to the Author

Reviewer #1: This is the first round of revision for the manuscript PONE-D-23-43692, entitled "ASV vs OTUs clustering: effects on alpha, beta and gamma diversities in microbiome metabarcoding studies", by Fasolo A and colleagues. In this paper the Authors try to bring their contribution to the ongoing discussion regarding the opportunity or not to work with ASV instead of OTUS in metabarcoding studies, showing a work focusing on the bacterial fraction of coastal environments.

The design of the experiments seems sound, and the manuscript is supported by adequate illustrations, though not very attractive, and supporting materials. Nonetheless, the work shows quite serious weaknesses, which need to be addressed before the manuscript can be accepted.

ANSWER: Anticipating what will be described below, as regards the illustrations, Fig. 3 has been implemented and two novel color figures (Fig.4 and Fig.5) have been added to show the results of the new analyses.

1. In light of the fact that, as mentioned, there is a fervent debate on whether or not to use ASV in numerical ecology studies, the work necessarily needs to be better framed. On the Fungal side, therefore in relation to the ITS marker, there are milestone works that support the validity of the use of OTUs (Kauserud 2023, Tedersoo et al. 2022), due to intraspecific variability in the ITS region, and sometimes intragenomic variability. The Authors must state much more clearly that their work refers to studies on bacterial communities, and introduce/discuss their arguments in more detail by referring to the opposite findings shown by the two works cited, both in the introduction and in the discussion.

[ https://www.sciencedirect.com/science/article/pii/S175450482300051X?via%3Dihub ]

[ https://onlinelibrary.wiley.com/doi/10.1111/mec.16460 ]

ANSWER: We thank the reviewer for pointing out this relevant aspect which we had not adequately emphasized, having mostly dealt with prokaryotic metabarcoding. We have added new text in the introduction stating the following and citing the two key references signaled as follows: 

“Within this rationale the ASV-based approach was developed to pursue a process which, at least in principle, starts as the opposite of the clustering. Rather than blurring reads into an averaging consensus, the method aims at focusing straight on exact sequences, (which, more realistically means, with the minimum possible reads agglomeration compromise)” 

“The ASV approach has nevertheless been critically scrutinized by authors that have remarked how its output is not to be misunderstood as that of truly unique and single sequences. This is because variants generated by the DADA2 step are actually stemming from a process that has also a low but inevitable degree of agglomeration and, strictly speaking, should therefore be regarded simply as a less-clustered type of output, but still on the conceptual continuum of the OTUs themselves (Kauserud, 2023). In addition, while the use of ASV has been essentially advocated for bacterial community studies (16S rDNA), when it comes to fungi and the target amplicon is the ITS, different issues as intraspecific and even intragenomic variability for repeated copies of the spacer have led to a reversed appreciation of these tools, showing that, for those eukaryotes, OTUs outperform ASV in resolving fungal diversity (Tedersoo et al, 2022), “ 

The aspect has been kept in consideration also in revising several other sentences throughout ye manuscript.

2. Line 68. I agree with this statement, but the problem is also how informative the molecular reference target is. It is also true that if we extend the discussion to other microorganisms we must take into account that when reasoning in terms of identity it sometimes happens that the "intra" variability exceeds the "inter" variability and only a phylogenetic approach can help to reconsider the placement.

ANSWER: We are indeed on the same line of thought. We have added a consideration (“In this respect, it can be foreseen that with the increase of the sequencing technologies throughput and of the ensuing bioinformatics, the basis for taxonomical assignments will be shifted from the metabarcoded amplicons to the Metagenome-Assembled Genomes (MAGs).”

3. The methods section is rather superficial, in particular regarding the bioinformatic analyses that follow the taxonomic assignment: there is no mention of the tools used for the calculation of the indices and other statistics, whether included in Qiime or not . Nowadays it is mandatory to provide all the information so that a certain analysis is repeatable; in this regard, I also suggest sharing the analytical workflow and the code used on github or other code sharing platform.

ANSWER: The section has been implemented and a two-page new Supplementary Information file (S1 Text) has been also added listing the command scripts of the entire bioinformatics pipeline scheme.

Moreover, as regards the indices and other statistics, including a number of newly added analyses, the revisedM&M paragraph of the bioinformatics includes the following updated information: 

“The ecological indices of Shannon-Wiener H value, Simpson’s 1-D, Community evenness (e^H/S), Brillouin, Menhinick, Margalef, Equitability J, Fisher alpha, and Berger-Parker, as well as Centered Log Ratio (CLR) transformation, Neighbor-Joining dendrograms based on Jaccard Distances or on Bray-Curtis Dissimilarity, the Pearson and Spearman Correlation and the Linear Discriminant Analysis (LDA) Effect Size to extract significantly differentially featured taxa, were all computed from the output data matrix using the Past 4.13 software [30]. LDA score values were calculated from the trimmed mean M values (TMM)-transformed data. Alpha-diversity and evenness significance of differences were estimated using the MicrobiomeAnalyst online utility (https://www.microbiomeanalyst.ca/ ). Principal Coordinate analysis and PERMANOVA for beta diversity assessment testing the following distance metrics: Binomial, Canberra, Clark, Raup, Wavehedges, were performed using the SHAMAN online utility (https://shaman.pasteur.fr ).” 

Furthermore, the missing information includes any method of normalization of the data or, alternatively, the justification for why it was not carried out (e.g. rarefaction or other methods). This is particularly important for understanding the indices and their ecological value/significance.

ANSWER: The details have been added: “As regards data transformation, for the analyses performed with the PAST software, data normalization using the Aitchison’s centered log-ratio transformation (CLR) was carried out to circumvent compositional dataset constraints. For the PCoA and PERMANOVA analyses, the Weighted Non-Null transformation option of the SHAMAN Suite was selected, while for the LDA carried out by the Microbiome Analyst suite either the CLR (in the PAST Software) or the Trimmed Mean M values (TMM) transformation (in the SHAMAN Suite) were compared”. 

4. In materials and methods, the description of the samples needs to be improved: as a reader I should not be forced to search previous work to obtain information about the nature of the samples. I have no idea, for example, what the fpi sample is. Looking at the S1 table, it seems that some samples comes in replicate, some others not: why?

ANSWER: The complete indication of each sample name acronym has been added to the legend of the new version of Fig.3. The sampling scheme design was that of the previously published reference [28] (from which the present report only used the existing raw data) and in that campaign the original experimental design had planned 9 areas with three replicates per sample, to which 8 extra samples, as single points were added to extend the transect to the seaside. In any event, the present manuscript is not aiming at quantifying microbiological/ecological relationships (existing in vivo) among the different habitat types of that landscape gradient as her we focus only on bioinformatical/numerical inferences (that arise in silico) from three different ways to process the same FASTQ raw sequence data of the digital archive stemming from that piece of research and deposited as NCBI SRA code PRJNA608631. 

There is no clearly stated sampling date (2016?)

ANSWER: The date has now been also specified in the appropriate Materials and Methods section. 

Raw sequencing data values are missing, as well as the final size of the samples after reads cleaning.

ANSWER: The information has been added “The sequencing had yielded 5.886.696 raw reads from which, after the filtering/denoising steps 2.579.452 were retained, 1.315.432 of which could be assigned to taxonomic identities.” 

Fig. 1 must be modified to report both clustering values of the OTUs.

ANSWER: The modification has been done and a new version of Fig.1 created.

5. Isn't the fact that alpha diversity indices drop simply a consequence inherent in the clustering of OTUs, which inevitably leads to fewer OTUs than ASVs and therefore to a reduction in index values?

ANSWER: This is correct, and it is indeed the inherently automatic portion of the effects of merging objects to create fewer sets. But the central aim of this work is in fact to point out that within this expected trend, the actual loss of diversity extent is concealed and can pass vastly overlooked until one quantifies it; e.g. if in a set of 1000 sequences there were 100 actually different variants but they would form only 10 groups sharing within each 97% identity, the resulting number of clustered OTUs would be 10. But if the 1000 sequences had been instead all different (1000 actual variants) but those differences would still occur within the ten 97%-coherent packages, the OTU-based inference could in that case underestimate by tenfold the actual genetic biodiversity of the sample. More explicitly from the numerical point of view, even with reads as short as 100 nucleotides, an OUT clustered at 97% cutoff will have 3 nucleotides free for variation included in the packaged set. The possible combinations accounted by four possible nucleotides distributed in three positions is 4^3= 64 combinations. Therefore, in those reads there is theoretically room for up to a 64-fold underestimation of the hidden diversity when adopting that common extent of clustering, Considering that actual NGS reads are more realistically twice as long, the resulting possibility for variability in six positions is 4^6= 4096 variants.

We have exemplified such aspects in the text to better clarify the principle. 

6. Is the phylogenetic tree in Fig.3 the best way to represent the effects on beta diversity? The Authors could delve deeper and explain better how and how much the ordination (b-diversity) changes, and better explain how the cluster analysis varies post-ordination (ASV, 97 and 99%), e.g. Which of the 3 methods generates clusters more consistent with multivariate orderings?

ANSWER: We thoroughly addressed these aspects and did the following. First we added a second series of distance trees in the new version of Fig.3 considering to compare also the effect of data transformation using the Aitchison’s centered log-ratio transformation (CLR) was to avoid as much as possible the biases from dataset compositionality, and used a distance metrics (Jaccard) less prone to that effect, and we compared the effects on NJ tree topology consistency with those seen upon calculating Bray-Curtis Dissimilarities on primary data. A further approach to visualize effects on beta diversity was to construct Principal Coordinate Analysis (PCoA) ordination biplots based on different distance metrics (Binomial, Canberra, Clark, Raup, Wavehedges) with each of the three dataset matrixes, and running a Permutational ANOVA (PERMANOVA) to inspect the significance of the differences (which yielded the new Tab 2) in which the beta-diversity was calculated among seven ecologically coherent sets across the land-to-sea transect (Cropped, Prairie, Hedges, Floodplain Transition, Coastal, Waters) within which the 34 samples were grouped. The three biplots obtained by the Raup distance are shown in the new Fig. 4.

Moreover we calculated the Linear Discriminant Analysis Effect Size evidencing the significantly (p<0.05) differentially featured taxa from each of the three dataset matrixes, their LDA scores and the up- or-down representation of each taxon within the seven macrohabitats. These new data originated the newly added Fig.5. 

7. The introduction at line 335 of the 'Valle Vecchia Oasis' is misleading, and should be done earlier.

ANSWER: The location first description and name was correctly anticipated at the beginning of the M&M section.

8. I believe that reporting the results of the gamma-diversity only to the entire area and the absence of a graphical representation is limiting, and makes this part of the results unattractive.

ANSWER: Being the gamma diversity defined as the total diversity in a landscape, if we get this query correctly, the graphical representation of the entire area intended is the view of the sampling location and/or that of the data in a tabulated fashion, and for such purpose we have added both a new supplementary figure (S1) and the new Tab.3.

9. The results are discussed as if they were generally applicable in metabarcoding studies, favoring the use of ASV more, but this has been shown not to apply for fungi and when using ITS as a target. The entire discussion must be reviewed with the awareness of having obtained contrasting results, highlighting the genomic causes that may have lead to these different results for Prokaryoti and Eukaryoti.

ANSWER: Capitalizing to what was anticipated above in responding to query n.1 and in the introduction in response to the opposite situation that occurs in fungal research, we have likewise revised this aspect throughout the manuscript.

For these reasons, I recommend major revision.

Reviewer #2: Fasolo et. al assessed the effect of different clustering methods (OTU vs ASV) on the diversity measures for microbiome data. The authors applied an amplicon-based metagenomics approach on samples collected from different habitats along a transect of 700 m and assessed the different clustering methods and different diversity measures on them. Among the main findings that the authors portray in the manuscript is the underestimation of the ecological indicators using reference based OTU clustering, therefore favoring the use of ASVs over OTU clustering for analyzing microbiome data. Although the idea of the manuscript is not totally novel and some similar papers have been published in recent years as Chiarello et al. 2022, Jeske and Gallert 2022 and Joos et al. 2020. I like the idea to increase the knowledge on this topic and evaluated this kind of methods in many environments as possible. However, I have some comments and some concerns in the way that data was analyzed that I will proceed to describe next:

- The authors describe in lines 153-155 that in their previous publication there is a detailed description of how diversity was estimated, but I would expect at least a brief description of software, packages and the utilized parameters because not everyone might be interested in your other paper. Moreover, the previous paper of the authors does not contain any description of the diversity measures portrayed in the current manuscript on github or other code sharing platform.

ANSWER: As anticipated also in reply to the same comment to Reviewer 1, the section has been implemented and a two-page new Supplementary Information file (S1 Text) has been also added listing the command scripts of the entire bioinformatics pipeline scheme 

As regards the diversity measures and other statistics, including a number of new added analyses The updated M&M paragraph of the bioinformatics includes the following updated information: 

“The ecological indices of Shannon-Wiener H value, Simpson’s 1-D, Community evenness (e^H/S), Brillouin, Menhinick, Margalef, Equitability J, Fisher alpha, and Berger-Parker, as well as Centered Log Ratio (CLR) transformation, 

---

## [Decision Letter · Decision Letter 1]

6 Aug 2024

ASV vs OTUs clustering: effects on alpha, beta and gamma diversities in microbiome metabarcoding studies

PONE-D-23-43692R1

Dear Dr. Squartini,

We’re pleased to inform you that your manuscript has been judged scientifically suitable for publication and will be formally accepted for publication once it meets all outstanding technical requirements.

Kind regards,

Federico Vita

Academic Editor

PLOS ONE

Additional Editor Comments (optional):

Dear authors, the revised process has been completed. Both reviewers agreed that the manuscript quality had been improved due to text revision, and it can be accepted for publication. Congratulations!

Reviewers' comments:

Reviewer's Responses to Questions

**Comments to the Author**

1. If the authors have adequately addressed your comments raised in a previous round of review and you feel that this manuscript is now acceptable for publication, you may indicate that here to bypass the “Comments to the Author” section, enter your conflict of interest statement in the “Confidential to Editor” section, and submit your "Accept" recommendation.

Reviewer #1: All comments have been addressed

Reviewer #2: All comments have been addressed

2. Is the manuscript technically sound, and do the data support the conclusions?

Reviewer #1: Yes

Reviewer #2: Yes

3. Has the statistical analysis been performed appropriately and rigorously? 

Reviewer #1: Yes

Reviewer #2: Yes

4. Have the authors made all data underlying the findings in their manuscript fully available?

Reviewer #1: Yes

Reviewer #2: Yes

5. Is the manuscript presented in an intelligible fashion and written in standard English?

Reviewer #1: Yes

Reviewer #2: Yes

6. Review Comments to the Author

Reviewer #1: This is the second revision round for the manuscript PONE-D-23-43692, entitled "ASV vs OTUs clustering: effects on alpha, beta and gamma diversities in microbiome metabarcoding studies", by Fasolo A and colleagues.

The manuscript has certainly benefited from the reviewers' comments, gaining clarity and usability.

The Authors have fully understood my notes and responded adequately, also implementing explanatory illustrations.

I will only point out a couple of things:

line 218, primer sequences in uppercase.

line 231, check correspondence with supplementary information numbering. I suggest using progressive numbering for supplementary material, in order to avoid attribution errors throughout the text.

Reviewer #2: Fasolo et. al have addressed properly the issues indicated during the first review, specially those regarding the details on the bioinformatics tools used to performing the research and most importantly the issue with data compositionality of microbiome dataset. The authors implemented proper tools and improved significantly the quality of the manuscript, making it to my view suitable to be publish in this journal.

I have only some minor comments listed next:

- Line 215 and/or in line 218 include the reference of the primers used for sequencing.

- Line 454, Initial T is in bold formatting.

- Line 498, remove the dot at the of the line.

7. PLOS authors have the option to publish the peer review history of their article (what does this mean?). If published, this will include your full peer review and any attached files.

Reviewer #1: No

Reviewer #2: No

---

## [Editor Report · Acceptance letter]

12 Aug 2024

PONE-D-23-43692R1 

PLOS ONE

Dear Dr. Squartini, 

I'm pleased to inform you that your manuscript has been deemed suitable for publication in PLOS ONE. Congratulations! Your manuscript is now being handed over to our production team.

Kind regards, 

on behalf of

Dr. Federico Vita 

Academic Editor

PLOS ONE